# In Vitro Antiviral and Virucidal Activity of Ozone against Feline Calicivirus

**DOI:** 10.3390/ani14050682

**Published:** 2024-02-22

**Authors:** Cristiana Catella, Francesco Pellegrini, Alice Carbonari, Matteo Burgio, Giovanni Patruno, Annalisa Rizzo, Claudia Maria Trombetta, Jolanda Palmisani, Vito Martella, Michele Camero, Gianvito Lanave

**Affiliations:** 1Department of Veterinary Medicine, University of Bari Aldo Moro, 70010 Valenzano, Italy; cristiana.catella@uniba.it (C.C.); francesco.pellegrini@uniba.it (F.P.); alice.carbonari@uniba.it (A.C.); matteo.burgio@uniba.it (M.B.); giovanni.patruno1@uniba.it (G.P.); annalisa.rizzo@uniba.it (A.R.); vito.martella@uniba.it (V.M.); michele.camero@uniba.it (M.C.); 2Department of Molecular and Developmental Medicine, University of Siena, 53100 Siena, Italy; trombetta@unisi.it; 3Department of Biosciences, Biotechnologies and Environment, University of Bari Aldo Moro, 70126 Bari, Italy; jolanda.palmisani@uniba.it

**Keywords:** ozone, feline calicivirus, norovirus, surrogate, in vitro, virucidal activity

## Abstract

**Simple Summary:**

Feline calicivirus (FCV) is a common pathogen of cats, displaying high contagiousness and resistance to many disinfectants. FCV infection can cause even fatal disease in cats. The virucidal efficacy of ozone (O_3_) has also been reported on naked viruses. In this study, the in vitro virucidal and antiviral activities of an ozone/oxygen (O_3_/O_2_) gaseous mixture were assessed against FCV. The antiviral activity of O_3_ was evaluated by exposing the virus to non-cytotoxic concentrations of the gaseous mixture. When confluent monolayers of CRFK cells were treated with the gas mixture after infection with FCV at a concentration of 50 μg/mL for 90 s, significant antiviral activity was observed with a decrease in viral titer of 1.75 log_10_ TCID_50_/50 μL. Virucidal activity was evaluated by exposing FCV to different concentrations (20, 35, and 50 μg/mL) of the gaseous mixture at distinct contact times, and a reduction in the viral titer by up to 2.25 log_10_ TCID_50_/50 μL was detected. The data obtained pave the way to the use of O_3_ as a disinfectant in cat environments at high risk of FCV transmission. Future studies will aim to assess the translational application of ozonation in disinfection of the food and beverage industry environments against human norovirus, which shares several biological similarities with FCV.

**Abstract:**

The *Caliciviridae* family includes several viral pathogens of humans and animals, including norovirus (NoV), genus *Norovirus*, and feline calicivirus (FCV), genus *Vesivirus*. Due to their resistance in the environment, NoV and FCV may give rise to nosocomial infections, and indirect transmission plays a major role in their diffusion in susceptible populations. A pillar of the control of viruses resistant to an environment is the adoption of prophylaR1.6ctic measures, including disinfection. Since NoVs are not cultivatable in common cell cultures, FCV has been largely used as a surrogate of NoV for the assessment of effective disinfectants. Ozone (O_3_), a molecule with strong oxidizing properties, has shown strong microbicidal activity on bacteria, fungi, protozoa, and viruses. In this study, the virucidal and antiviral activities of an O_3_/O_2_ gas mixture containing O_3_ were tested at different concentrations (20, 35, and 50 μg/mL) for distinct contact times against FCV. The O_3_/O_2_ gas mixture showed virucidal and antiviral activities against FCV in a dose- and contact time-dependent fashion. Ozonation could be considered as a valid strategy for the disinfection of environments at risk of contamination by FCV and NoV.

## 1. Introduction

Ozone (O_3_), a molecule consisting of three oxygen atoms, has attracted the attention of medical practitioners for its powerful oxidizing properties, which can disrupt the structure of microorganisms and induce their inactivation [1,2,3]. The application of O_3_ in medical settings is an existing research topic, although concerns regarding its safety and efficacy are still debated.

Due to its oxidizing properties, O_3_ is suitable for multiple purposes, ranging from specific medical therapies to surface disinfection [4,5,6]. O_3_ dosage and exposure time should be precisely assessed to ensure treatment effectiveness and to avoid adverse effects on materials or patients exposed to the substance [7].

High levels of O_3_ ranging between 20 and 500 ng/mL have been proven to damage neuronal cell cultures during in vitro tests [8]; moreover, direct exposure to O_3_ irritates the respiratory system and exacerbates pre-existing inflammatory conditions in the lungs [9].

O_3_ generators have been used for disinfecting indoor air and neutralizing viruses, i.e., influenza virus and coronavirus, including SARS-CoV-2 [10,11]. O_3_ is able to reduce viral infectivity in laboratory settings, including against herpesvirus [3]. O_3_ has been successfully used in water treatment plants to disinfect wastewater, causing a reduction in viral loads [12]. The treatment of water with O_3_ effectively inactivates picornavirus and norovirus (NoV), making the treated water suitable for the disinfection of healthcare settings and food preparation areas [13,14].

Furthermore, managers of animal shelters, farms, and veterinary practitioners have shown interest in ozone generators due to their effectiveness in disinfection against a wide range of enveloped and non-enveloped viral pathogens, i.e., feline and canine coronaviruses, feline calicivirus (FCV), and feline panleukopenia parvovirus [15].

O_3_ dissolved in water can affect the infectivity of different microorganisms, including viruses, and can trigger the oxidation of organic and inorganic substances (i.e., pesticides, pharmaceuticals, and organic matter) without generating hazardous by-products, thus improving the overall water quality [16,17].

Despite the promising results observed with the in vitro use of O_3_, translation of these results to practice in healthcare or public settings requires the careful evaluation of safety and efficacy [18].

The *Caliciviridae* family includes several viral pathogens of humans and animals [19], including NoVs (genus *Norovirus*), regarded as major etiological agents of acute gastroenteritis (AGE) in human population [20,21]. NoVs are associated with AGE in the elderly and newborns, and are also known as “winter disease” [22,23,24,25]. Human NoVs (HNoVs) cause approximately 200,000 deaths per year worldwide, and a major modality of virus transmission is represented by the fecal–oral route [21]. The prevention of NoV infection requires the disinfection of water, contaminated surfaces, and the hands of operators handling food. Many studies on the inactivation of NoVs have been carried out on different matrices with different results [26,27].

Feline calicivirus (FCV), genus *Vesivirus*, is a major pathogen of cats, associated with a mild-to-severe, highly contagious disease in cats. The disease is common in shelters and breeding colonies and often infects young cats. Also, due to their resistance in the environment, FCV may give rise to nosocomial infections in veterinary hospitals [28]. Since the cultivation of human NoVs is difficult, FCV is often used as a surrogate of NoVs due to their similarities in terms of genome organization, replication, and resistance to chemical agents [29].

The development of novel strategies for the disinfection of surfaces and matrices is relevant in terms of animal health in the management of infectious diseases of animals and also in terms of human health. In this study, an O_3_/O_2_ gas mixture was tested against FCV in vitro at different concentrations and for different time intervals.

## 2. Materials and Methods

### 2.1. O_3_ Generator

O_3_ generators are medical devices able to produce an O_3_/oxygen (O_3_/O_2_) gas mixture by means of either extremely high electrical voltages or UV radiation; the process breaks the connection between O_2_ molecules into O_2_ atoms, which in proximity of an excess amount of O_2_ molecules form the three-atom O_3_ molecule. After connection to an electrical source and an O_2_ cylinder, the medical device used in our study (Vet-Ozone Medica srl-Italia, Bologna, Italy) converts the O_2_ (substrate) into O_3_ by electrical discharges, thus producing an O_3_/O_2_ gas mixture containing different concentrations of O_3_ (20, 35, and 50 μg/mL).

### 2.2. Hermetic Box for Gas Flow

A container was handcrafted to expose the Petri dishes to the O_3_ flow, as previously described [3].

### 2.3. Cells and Viruses

Crandell Reese Feline Kidney (CRFK) cells were grown at 37 °C in a 5% carbon dioxide (CO_2_) atmosphere in Dulbecco Minimal Essential Medium (D-MEM) supplemented with 10% fetal bovine serum, 100 IU/mL penicillin, 0.1 mg/mL streptomycin, and 2 mM l-glutamine. The same medium was used for antiviral tests. The FCV strain (F9) was cultured and titrated in CRFK cells. The viral stock with a titer of 6.25 log_10_ tissue culture infectious dose (TCID_50_)/50 μL was stored at −80 °C and used for the experiments.

### 2.4. Cytotoxicity Assay

Cytotoxicity test was carried out to establish the concentration and contact time of the O_3_/O_2_ gas mixture on the CRFK cells. Confluent monolayers of CRFK cells, grown for 24 h in 35 mm Petri dishes and maintained in D-MEM, were exposed to the O_3_/O_2_ gas mixture at different O_3_ concentrations (20, 35, and 50 μg/mL) at room temperature for different contact times: 30, 60, 90, 180, and 300 s. Negative controls were set up on confluent monolayers of CRFK cells, cultured for 24 h in 35 mm Petri dishes, and maintained in D-MEM without O_3_/O_2_ delivery, maintaining the same temperature and contact times. Cytotoxicity was assessed through the direct microscopic examination of cell morphology (loss of cell monolayer, granulation, cytoplasmic vacuolization, stretching and shrinking of cell extensions, and darkening of cell borders) [30]. Moreover, indirect measurement of cell viability were obtained by means of an in vitro toxicological analysis by using a kit (Sigma–Aldrich Srl, Milan, Italy) based on 3-(4,5-dimethylthiazol-2yl)-2,5-diphenyltetrazolium bromide (XTT). The XTT test was performed as previously described [30]. The optical density (OD) values were assessed to calculate the percentage of cytotoxicity (percentage of dead cells) according to the following formula: % Cytotoxicity = [(OD_control cells_ − OD_treated cells_) × 100]/OD_control cells_. The tests were performed in triplicate and data were expressed as mean ± SD. Exposure conditions that did not reduce the viability of treated CRFK cells by more than 20% (cytotoxicity threshold) were considered non-cytotoxic and were selected for subsequent antiviral testing.

### 2.5. Antiviral Assays

Based on the cytotoxicity assay results, the antiviral activity against the FCV was evaluated by using the O_3_/O_2_ gas mixture containing O_3_ at 20, 35, and 50 μg/mL for 30, 60, and 90 s. To assess the pathway of viral inhibition against FCV induced by the O_3_/O_2_ gas mixture containing O_3_, two different protocols (A and B) were carried out, as is detailed below. All experiments were performed in triplicate.

#### 2.5.1. Protocol A: Virus Infection of Cell Monolayers before Treatment with O_3_

Confluent monolayers of CRFK cells grown for 24 h were used in 24-well plates. Cell monolayers were infected with 100 μL of viral suspension containing 100 TCID_50_ FCV. The virus was adsorbed for 1 h at 37 °C and subsequently removed, the cell monolayers were washed once with D-MEM, and subsequently D-MEM (1 mL) was added. Cell monolayers in three independent 24-well plates were treated with the O_3_/O_2_ gas mixtures containing O_3_ at 20, 35, and 50 μg/mL for 30, 60, and 90 s. Untreated infected cell monolayers were used as a control virus (CV). After 72 h, aliquots of the supernatants were collected for subsequent viral titration.

#### 2.5.2. Protocol B: Viral Infection of Cell Monolayers after Treatment with O_3_

Confluent monolayers of CRFK cells grown for 24 h were used in 24-well plates. Cell monolayers in three independent 24-well plates were treated with O_3_/O_2_ gas mixtures containing O_3_ at 20, 35, and 50 μg/mL for 30, 60, and 90 s. Subsequently, the monolayers were washed once with D-MEM and infected with 100 μL of viral suspension containing 100 TCID_50_ FCV. After virus adsorption for 1 h at 37 °C, the inoculum was removed, the monolayers were washed once with D-MEM, and subsequently D-MEM (1 mL) was added. Untreated infected cell monolayers were used as the CV. After 72 h, aliquots of each supernatant were collected for subsequent viral titration.

### 2.6. Virucidal Activity Assay

The virucidal activity of O_3_ against FCV was evaluated using O_3_/O_2_ gas mixtures containing O_3_ at 20, 35, and 50 μg/mL. One ml of FCV stock virus was posed into 35 mm Petri dishes and directly exposed to the O_3_/O_2_ gas mixture in the hermetic box at room temperature. At different contact times (30, 60, 90, 180, 300, and 600 s), 100 µL of the treated viral suspension was collected for subsequent viral titration. An untread aliquot of FCV stock virus (1 mL) was used as the CV, maintained at room temperature, and collected at different contact times (30, 60, 90, 180, 300, and 600 s) for viral titration. The experiments were performed in triplicate.

### 2.7. Viral Titration

Ten-fold dilutions (up to 10^−8^) of each supernatant were titrated in quadruplicates in 96-well plates containing CRFK cells. The plates were incubated for 72 h at 37 °C in 5% CO_2_. The cytopathic effect of FCV on the CRFK cells was evaluated using an inverted microscope with live-cell imaging. Based on the cytopathic effect, TCID_50_/50 μL was calculated according to the Reed–Muench method [31].

### 2.8. Data Analysis

All data were expressed as mean ± SD and analyzed using the GraphPad Prism (v 9.5.0) program (Intuitive Software for Science, San Diego, CA, USA). To assess the normality of distribution, the Shapiro–Wilk test was performed. Two-way factorial ANOVA with O_3_ concentration of the O_3_/O_2_ gas mixture and contact times as factors as well as the Tukey test as a post hoc test were applied to cytotoxicity results. One way ANOVA tests were performed on the results of the virucidal and antiviral activities at different contact times, considering the fixed O_3_ concentration of the O_3_/O_2_ gas mixture. Statistical significance was set at 0.05.

## 3. Results

### 3.1. Cytotoxicity Assay

Direct exposure of CRFK cells to O_3_/O_2_ gas mixtures containing O_3_ at 20, 35 and 50 μg/mL did not produce any changes in cell morphology at 30, 60, and 90 s (Figure 1A), whereas cytotoxicity effects were consistently observed at the subsequent contact times (180 and 300 s) (Figure 1B,C).

Morphological observations overlapped with cytotoxicity results using the XTT test. Cell exposure to O_3_/O_2_ gas mixtures containing O_3_ at 20, 35 and 50 μg/mL at different time intervals (30, 60, 90, 180, and 300 s) resulted in a cytotoxicity increase in a dose- and contact time-dependent fashion (Figure 2).

In detail, the O_3_/O_2_ gas mixture containing O_3_ at 20 μg/mL at 30 and 60 s induced mean cytotoxicity values of 2.0% (SD ± 0.15), whilst at 90 s the mean cytotoxicity was 3.7% (SD ± 1.1) below the cytotoxic threshold. Mean cytotoxicity values of 21.5% (SD ± 1.2) and 82.1 (SD ± 2.2) were observed at 180 and 300 s, respectively (Figure 2A). The O_3_/O_2_ gas mixture containing O_3_ at 35 μg/mL at 30, 60, and 90 s induced mean cytotoxicity values of 2.7% (SD ± 0.13), 3.5% (SD ± 0.95), and 3.8% (SD ± 1.1), respectively, below the cytotoxic threshold. Mean cytotoxicity values of 22.6% (SD ± 1.2) and 81.3% (SD ± 0.10) were observed at 180 and 300 s, respectively (Figure 2B). The O_3_/O_2_ gas mixture containing O_3_ at 50 μg/mL at 30, 60, and 90 s induced mean cytotoxicity values of 2.6% (SD ± 0.15), 2.6% (SD ± 0.17), and 4.4% (SD ± 1.1), respectively, below the cytotoxic threshold. Mean cytotoxicity values of 30.9% (SD ± 1.2) and 83.3 (SD ± 2.4) were observed at 180 and 300 s, respectively (Figure 2C).

### 3.2. Antiviral Activity Assay

#### 3.2.1. Protocol A: Treatment of Infected Cell Monolayers with O_3_

To understand if there are differences in O_3_ activity in the early stages of infection, we treated CRFK cells after 1 h absorption with FCV. In the comparison of the viral titer of CV with infected CRFK cells treated for 30, 60, and 90 s with O_3_/O_2_ gas mixtures containing O_3_ at 20 and 35 μg/mL, slight decreases in viral titer of up to 0.50 log_10_ TCID_50_/50 µL were observed, although without statistical significance (*p* > 0.05). Infected cells treated with the O_3_/O_2_ gas mixture containing O_3_ at 50 μg/mL for 30 and 60 s exhibited modest reductions in viral titer of up to 0.50 log_10_ TCID_50_/50 µL, also lacking statistical significance (*p* > 0.05) when compared to the viral titer of CV. By using the O_3_/O_2_ gas mixture containing O_3_ at 50 μg/mL for 90 s, a significant decline in viral titer of 1.75 log_10_ (*p* < 0.05) was induced in comparison to the CV (Figure 3).

#### 3.2.2. Protocol B: Treatment of Cell Monolayers with O_3_ before Virus Infection

To understand if treatment with O_3_ can alter/affect the receptor binding of FCV, we treated CRFK cells before absorption with FCV. Comparing the viral titer of the CV with infected cells treated with O_3_/O_2_ gas mixture containing O_3_ at 20, 35 and 50 μg/mL for 30, 60, and 90 s, limited reductions in viral titer of up to 0.75 log_10_ TCID_50_/50 µL were observed, although without statistical significance (*p* > 0.05) (Figure 3).

### 3.3. Virucidal Activity Assay

O_3_/O_2_ gas mixture containing O_3_ at 20 μg/mL significantly reduced FCV titers by 0.75 log_10_ TCID_50_/50 μL (*p* < 0.05) at 30 s and 60 s; by 1.00 log_10_ TCID_50_/50 μL (*p* < 0.05) at 90 s; by 1.25 log_10_ TCID_50_/50 μL (*p* < 0.05) at 180 s; by 1.75 log_10_ TCID_50_/50 μL (*p* < 0.05) at 300 s; and by 2.00 log_10_ TCID_50_/50 μL (*p* < 0.05) at 600 s when compared to the CV (Figure 4). O_3_/O_2_ gas mixture containing O_3_ at 35 μg/mL was able to significantly decrease FCV titers by 0.75 log_10_ TCID_50_/50 μL (*p* < 0.05) at 30 s and 60 s; by 1.00 log_10_ TCID_50_/50 μL (*p* < 0.05) at 90 s; by 1.25 log_10_ TCID_50_/50 μL (*p* < 0.05) at 180 s; by 1.75 log_10_ TCID_50_/50 μL (*p* < 0.05) at 300 s; and by 2.25 log_10_ TCID_50_/50 μL (*p* < 0.05) at 600 s when compared to the CV (Figure 4). O_3_/O_2_ gas mixture containing O_3_ at 50 μg/mL induced significant FCV titer reductions of 1.00 and 1.25 log_10_ TCID_50_/50 μL (*p* < 0.05) at 30 s and 60 s, respectively; of 1.50 log_10_ TCID_50_/50 μL (*p* < 0.05) at 90 s; of 1.5 log_10_ TCID_50_/50 μL (*p* < 0.05) at 180 s; of 2.00 log_10_ TCID_50_/50 μL (*p* < 0.05) at 300 s; and of 2.25 log_10_ TCID_50_/50 μL (*p* < 0.05) at 600 s when compared to the CV (Figure 4).

## 4. Discussion

Several beneficial effects of O_3_ are widely described in the literature, i.e., antibacterial, antifungal, and antiviral properties [2,3]. The anti-inflammatory, analgesic, immunomodulatory, and healing properties have also been reported [32].

In this study, the virucidal and antiviral activities of O_3_ were evaluated against FCV, a common and highly contagious pathogen of domestic cats resistant to many disinfectants and largely used as a surrogate for HNoV [33].

The antiviral activity of the O_3_/O_2_ gas mixture containing O_3_ was assessed at different O_3_ concentrations (20, 35, and 50 μg/mL) for three contact times (30, 60, and 90 s). The contact times were selected based on the cytotoxic activity assessed by the XTT test on CRFK cells for different time points (30 to 300 s). For the three tested O_3_ concentrations, time contacts at 30, 60, and 90 s were regarded as non-cytotoxic (below the cytotoxicity threshold of 20%). At later time contact points, starting from 180 s, an increment in cytotoxicity was observed primarily at the concentration of 50 μg/mL (over 30%). To understand if O_3_ has activity on FCV at the early stages of infection or if it can affect binding to FCV-specific receptors, we designed two different experiments in which CRFK cells were treated with O_3_ after (protocol A) or before (protocol B) FCV infection. In protocol A, O_3_ displayed more promising results in comparison to those observed in protocol B. The highest ozone concentration (50 μg/mL) used against FCV induced a significant decrease (1.75 log_10_ TCID_50_/50 μL) in viral titer after a 90 s exposure, thus indicating a potential dose and contact time-dependent anti-replicative effect of O_3_ against FCV.

The virucidal activity of the O_3_/O_2_ gas mixture was also evaluated using O_3_ at different concentrations (20, 35, and 50 μg/mL) for six contact times (30, 60, 90, 180, 300, and 600 s), with the latter three being over the cytotoxic threshold. Consistently significant decreases in viral titer up to 2.25 log_10_ TCID_50_/50 μL were observed when using the three O_3_ concentrations for all the contact times in comparison to the CV.

The results of in vitro virucidal and antiviral activities obtained in this study underline the importance of the use of higher O_3_ concentrations and longer exposure times of the compound against FCV (Figure 3 and Figure 4). Moreover, our study confirms the efficacy of O_3_ as a disinfectant. The ozonation treatment against non-enveloped viruses was proven to be able to induce oxidation of the viral capsid proteins, thus preventing viral infection of the sensitive cells through either capsid destruction or inability to bind to the cellular receptor [34].

In human medicine, the therapeutic treatment of NoV infection is based on relieving symptoms through oral or intravenous rehydration. The highly infectious nature of NoV, its stability in the environment, its long-term viral shedding, and the lack of a reproducible cultivation system lead to large gastroenteritis outbreaks and make the development of a control strategy a challenging problem [35]. Several antivirals have been tested in vitro against HNoV despite no drug being proven able to treat and/or prevent HNoV infections [36]. Also, several NoV vaccines are in development, including vaccines in preclinical trials, although to date no prophylactic vaccines are available on the market [37].

In veterinary medicine, the prevention of the FCV infection is carried out both through symptomatic therapy and vaccination [33]. Despite the good protection against FCV-associated acute oral and upper respiratory tract disease often provided by vaccination in cats, infection or post-infection FCV shedding is not prevented [38]. However, significantly lower infection rates are reported in vaccinated cats compared to unvaccinated cats [39,40].

The use of disinfection is pivotal to counteracting virus transmission and therefore could represent a powerful tool for disease prevention in both human and veterinary medicine. In order to stem NoV spread, sanitization and disinfection are used in the food industry to guarantee high standards of hygiene, thus reducing the risks of food-borne infections. Additionally, in cat shelters, pet boarding houses, breeding feline colonies, cat shows, and catteries have disinfection procedures that could play a pivotal role in preventing the spread of infectious agents such as FCV, to which sensitive cats are easily exposed.

Several studies have compared the antimicrobial activity of O_3_ on different surfaces and biological matrices with conventional disinfectants, i.e., sodium hypochlorite and chrlorexidine [41,42]. O_3_ has proven to be an energetic, interesting, alternative, and safe disinfectant with scarce and harmless environmental residues. Conversely, chlorination could leave residues toxic to humans and wildlife because its decomposition produces trihalomethanes and other halo-organic carcinogens [43]. O_3_, unlike sodium and calcium hypochlorite, does not induce corrosion on the steel components of water systems, healthcare facilities, and surfaces where food is processed and handled [44].

According to protocol nr 24,482 dated on 31 July 1996, the Italian Ministry of Health has recognized the use of O_3_ as a natural device for the sterilization of environments contaminated by bacteria, viruses, spores, molds, and mites and for the treatment of air and water. O_3_ does not harm food and has been approved both by the Italian Ministry of Health and by the Food and Drug Administration as a food preservative due to its antimicrobial efficacy.

The cost-effectiveness and rapid action of O_3_ in decreasing viral titers highlight its promising efficacy as an effective disinfection strategy against caliciviruses. Our study has several limitations. Other studies should assess the in vitro broad-spectrum antiviral and virucidal efficacy of O_3_, taking into account the genetic and phenotypic diversity of enteric viruses even below the species level. For instance, marked differences in terms of resistance to chemical and physical inactivation have been observed between NoV GII.4 variants [45,46,47,48]. Likewise, marked differences in terms of resistance phenotypes have been observed between enteric and respiratory FCV isolates [49]. Furthermore, research studies are needed to validate the effectiveness of O_3_ in practical scenarios, such as catteries and shelters.

These data open new perspectives for future applications of ozonation in human and veterinary medicine settings due to its low maintenance costs, high efficacy in the sanitation of air and surfaces of closed environments, and the easy availability of O_3_ generators.

## 5. Conclusions

In conclusion, we have demonstrated the in vitro virucidal and antiviral activities of O_3_ against FCV. Antiviral efficacy was displayed in the decreases in viral titers, which were dose- and contact time-dependent. The virucidal efficacy of O_3_ against FCV, observed at different concentrations and exposure times, highlights its potential use in mitigating FCV transmission, chiefly within cat shelters and catteries. The housing of cats in non-domestic environments may represent a significant risk due to the high contagiousness and persistence of FCV. Implementation of ozonation in the veterinary field as a disinfection strategy could represent a promising tool for the control of the disease.

The results of our study may also suggest the potential of O_3_ as a powerful disinfectant against HNoV in food and beverage handling, considering the irrelevant impact of ozonation on human health, ecosystems, and resources [44]. Due to its excellent oxidation and disinfection qualities, O_3_ is widely used for the treatment of drinking water, the removal of organic and inorganic matter, and the oxidation of several pesticides [17,50].

Further studies that focus on the optimization of ozone delivery systems are needed to define standardized disinfection protocols for indoor environments. These efforts could pave the way for the practical and efficient use of ozone as a proactive measure against FCV/HNoV outbreaks in human and veterinary settings.

## Figures and Tables

**Figure 1 animals-14-00682-f001:**
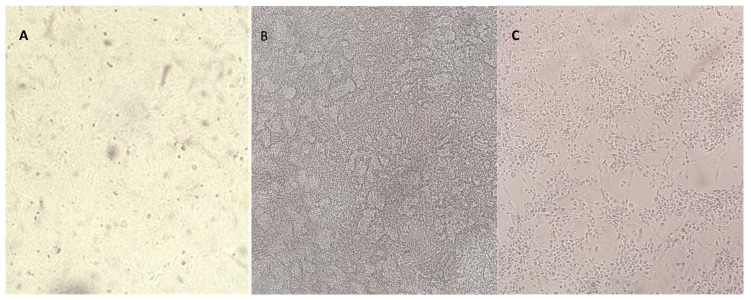
Cytotoxic effect of O_3_ at 50 μg/mL on 24 h monolayer of Crandell Reese Feline Kidney (CRFK) cells with live-cell imaging (magnification 10×) at 30–90 s (**A**), 180 s (**B**), and 300 s (**C**). Exposure of CRFK cells to O_3_ at lower concentrations produced similar results.

**Figure 2 animals-14-00682-f002:**
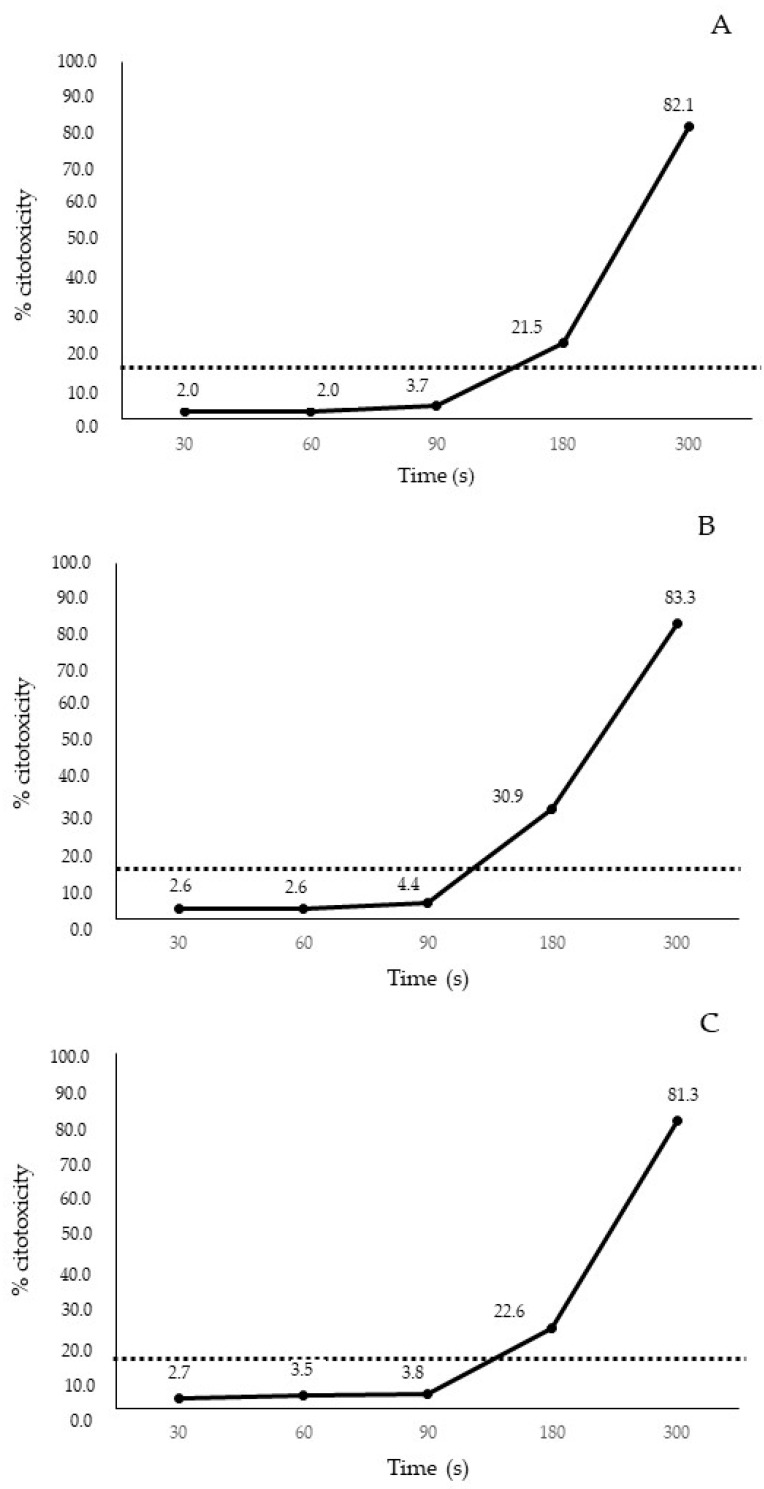
Cytotoxicity of CRFK cells (express as percentage) treated with O_3_/O_2_ gas mixture containing O_3_ at 20 μg/mL (**A**), 35 μg/mL (**B**), and 50 μg/mL (**C**) plotted against contact times. The horizontal dotted line indicates the cytotoxicity threshold of (20% of cell death).

**Figure 3 animals-14-00682-f003:**
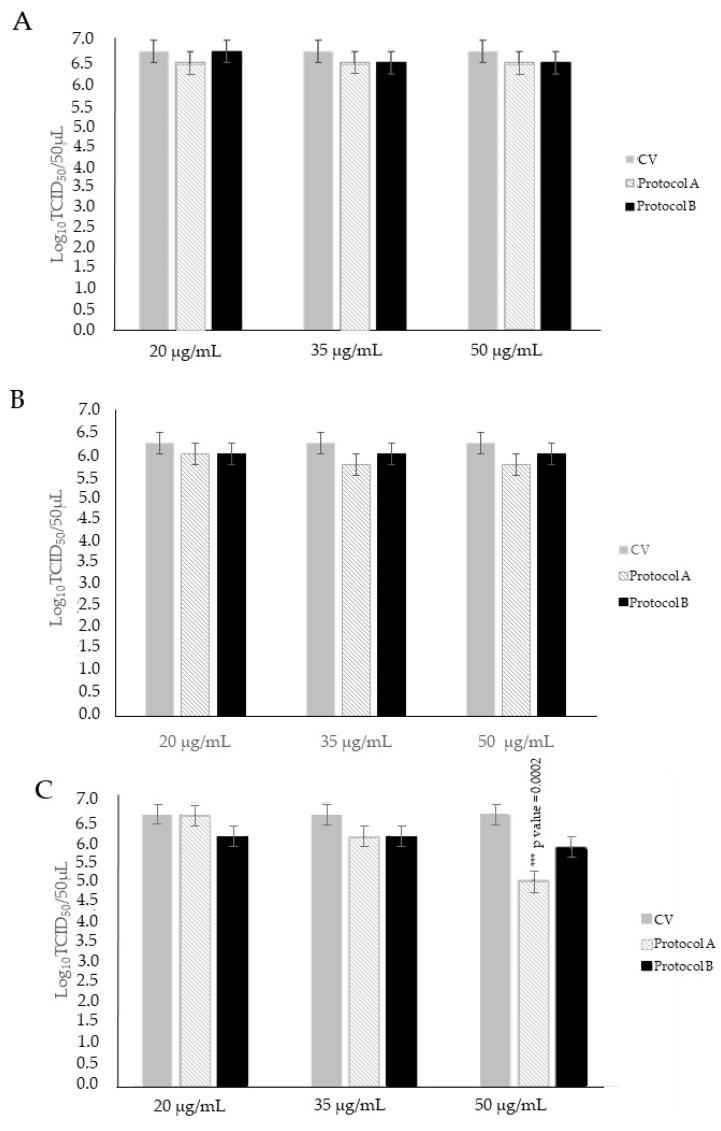
Antiviral activity of ozone (O_3_) at different concentrations (20, 35 and 50 μg/mL) against feline calicivirus (FCV). Treatment of infected cell monolayers with O_3_ (protocol A). Treatment of cell monolayers with O_3_ before virus infection (protocol B). FCV not treated (control virus, CV) and treated with O_3_ at room temperature for 30 s (**A**), 60 s, (**B**) and 90 s (**C**) were subsequently titrated on Crandell Reese Feline Kidney (CRFK) cells. Viral titers of FCV are expressed as log_10_ TCID_50_/50 μL. Significant *p* values are displayed. Bars in the figures indicate the means. Error bars indicate the standard deviation.

**Figure 4 animals-14-00682-f004:**
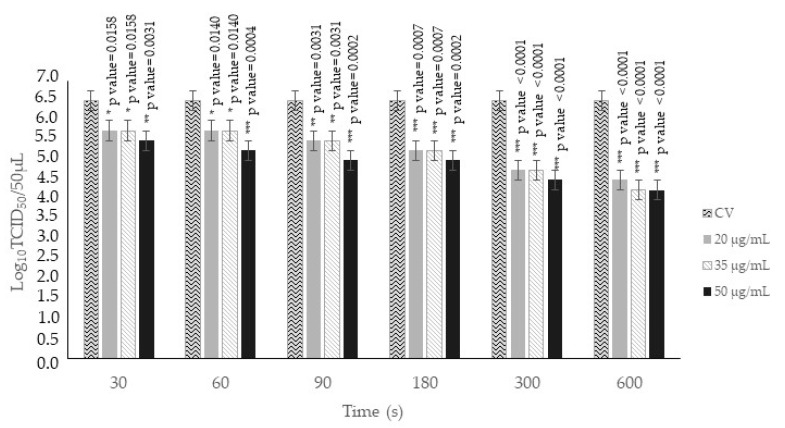
Virucidal activity of ozone (O_3_) at different concentrations (20, 35, and 50 μg/mL) against feline calicivirus (FCV). FCV was incubated with O_3_ for 30, 60, 90, 180, 300, and 600 s at room temperature. FCV not treated (control virus, CV) and treated with O_3_ were subsequently titrated on Crandell Rees Feline Kidney (CRFK) cells. Viral titers of FCV are expressed as log_10_ TCID_50_/50 μL. Significant *p* values are displayed. Bars in the figures indicate the means. Error bars indicate the standard deviation.

## Data Availability

The data that support the findings are contained in the paper.

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
