# Peer review of "In Vitro Antiviral and Virucidal Activity of Ozone against Feline Calicivirus"

_animals, 2024, doi:10.3390/ani14050682_

Round 1

Reviewer 1 Report

Comments and Suggestions for Authors

In this work, the authors tested the virucidal and antiviral activity of O3/O2 gas mixture against FCV. For this purpose, cells were exposed to different 03/02 concentrations and times before or after infection with FCV, while to test the virucidal effect, FCV was directly exposed to different 03/02 concentrations before infection. Virus titers were obtained from all conditions tested and compared with those for the untreated control conditions. 

It is a well-written paper with clear and statistically analyzed results.  

  • The title could include the antiviral and virucidal activity of Ozone against FCV.
  • Fig. 1 does not indicate the conditions of the image shown in Fig. 1B. It would be worth showing images of the conditions tested and described in lines 185-188.
  • Fig. 3 and 4. It would be helpful to show the p values (*) in the graphs.
  • Fig. 4 legend: Please revise.
  • Lines 317-319: Adding the virus before or after O3/O2 treatment does not allow identifying the stage of viral replication inhibited, but if it has to be or not with virus binding/entry. It would be worth mentioning it in the results section. Could this be related to the statement mentioned in lines 332-334? 

Line 15: is common pathogen of cats.

Line 54: Please indicate the range of O3 considered "high levels".

Line 202: Please indicate Fig. 2 at the end of the sentence.

Comments on the Quality of English Language

None

Author Response

Dear Referee,

herein you can find a point-by-point response to your comments for the manuscript animals-2826879 entitled “Activity of Ozone against Feline Calicivirus” submitted to Animals.

Thanks in advance

Best regards

Gianvito Lanave

In this work, the authors tested the virucidal and antiviral activity of O3/O2 gas mixture against FCV. For this purpose, cells were exposed to different O3/O2 concentrations and times before or after infection with FCV, while to test the virucidal effect, FCV was directly exposed to different O3/O2 concentrations before infection. Virus titers were obtained from all conditions tested and compared with those for the untreated control conditions. It is a well-written paper with clear and statistically analyzed results. 

R1.1. The title could include the antiviral and virucidal activity of Ozone against FCV.

Reply to R1.1. We agree with the referee for this suggestion. Accordingly, we modified the title as follows: “In vitro antiviral and virucidal activity of Ozone against Feline Calicivirus “.

R1.2. Fig. 1 does not indicate the conditions of the image shown in Fig. 1B. It would be worth showing images of the conditions tested and described in lines 185-188.

Reply to R1.2. We agree with the referee that Figure 1 could be more informative. Accordingly, we displayed in the figure the cytotoxic effects of O3 on CRFK cells at different time intervals (30-90 s, 180 s, and 300 s, panels A, B, and C, respectively). Figure 1 shows that O3 on CRFK cells displayed evident cytotoxic effects at 50 µg/ml only after 180 and 300s. The same results were observed at lower O3 concentrations.

R1.3. Fig. 3 and 4. It would be helpful to show the p values (*) in the graphs.

Reply to R1.3.  We added p-values in both figures, as requested.

R1.4. Fig. 4 legend: Please revise.

Reply to R1.4. Fig 4 legend was revised, as suggested.

R1.5. Lines 317-319: Adding the virus before or after O3/O2 treatment does not allow identifying the stage of viral replication inhibited, but if it has to be or not with virus binding/entry. It would be worth mentioning it in the results section. Could this be related to the statement mentioned in lines 332-334?

Reply to R1.5. We agree with the referee R1’s comment that this part may not be clear to the readers. We modified the heading of section 3.2.1 and 3.2.2. 

At section 3.2.1, (page 6 lines 246-248) the heading was changed to “Protocol A: treatment with O3 of infected cell monolayers”

and we added a sentence, as requested by R1 “To understand if there are differences in O3 activity at the early stages of infection we treated CRFK cells after 1h absorption with FCV.”

At section 3.2.2, (page 7 lines-257-259) the heading was changed to “Protocol B: treatment with O3 of cell monolayers before virus infection” and a sentence was added, following R1’s suggestion, “To understand if treatment with O3 can alter/affect the receptor binding of FCV, we treated CRFK cells before absorption with FCV.”

In the discussion, at page 9, line 333-336, we re-phrased a sentence “To understand if O3 has activity on FCV at the early stages of infection or it can affect binding to FCV-specific receptors, we designed two different experiments, in which CRFK cells were treated with O3 after (protocol A) or before (protocol B) FCV infection.

R1.6. Line 15: is a common pathogen of cats.

Reply to R1.6. This was corrected (line 16 page 1).

R1.7. Line 54: Please indicate the range of O3 considered "high levels".

Reply to R1.7. The information was added in the text, as requested. The levels mentioned in the reference # 8 (Ankul Singh et al., 2023), i.e. between 20 and 500 ng/ml, have been indicated (line 55 page 2).  

R1.8. Line 202: Please indicate Fig. 2 at the end of the sentence.

Reply to R1.8. This was corrected (line 203 page 5).

Reviewer 2 Report

Comments and Suggestions for Authors

1. Is the cytotoxicity detection method used in this paper a classic standardized method? If not, the process of establishing a cytotoxic assay should be described and standardized.

2. The killing effect of O3 on FCV of infected monolayer cells was tested in this paper. However, in practical applications, O3 is mostly used for environmental disinfection, including air, water and surfaces of objects. It is suggested that the authors use FCV-contaminated cattery to evaluate the efficacy of O3 in killing viruses, which is more useful than using FCV-infected monolayer cells in the laboratory.  

Author Response

Dear Referee,

herein you can find a point-by-point response to your comments for the manuscript animals-2826879 entitled “Activity of Ozone against Feline Calicivirus” submitted to Animals.

Thanks in advance

Best regards

Gianvito Lanave

R.2.1. Is the cytotoxicity detection method used in this paper a classic standardized method? If not, the process of establishing a cytotoxic assay should be described and standardized.

Reply to R.2.1.  The cytotoxicity detection method used in this paper represents a standardized method composed of direct microscopic examination of cell morphology and indirect measurement of cell viability obtained using an in vitro toxicological analysis kit, as described in the paper by Lanave et al., 2017 (https://doi.org/10.1016/j.jviromet.2017.07.012).

R.2.2. The killing effect of O3 on FCV of infected monolayer cells was tested in this paper. However, in practical applications, O3 is mostly used for environmental disinfection, including air, water and surfaces of objects. It is suggested that the authors use FCV-contaminated cattery to evaluate the efficacy of O3 in killing viruses, which is more useful than using FCV-infected monolayer cells in the laboratory.

Reply to R.2.2. We agree with the referee that O3 could be used for environmental applications. Our study in vitro is a preliminary step useful to assess if O3 can be applied in FCV-infected cattery. We mentioned this potential application in the discussion at page 9, line 370-372.

Reviewer 3 Report

Comments and Suggestions for Authors

FCV is a common highly contagious virus affecting cats, and as other viruses of the Caliciviridae family, is known to be resistant to many disinfectants. Catella et al. examined the impact of gaseous ozone on FCV viability in vitro using a hermetic box for gas flow.

The effect of gaseous ozone on disinfection of noroviruses has been previously studied, for example in the context of hospital settings (Hudson et al 2007, Journal of Hospital Infections) or food disinfection (Predmore et al 2015, Food Microbiology; Hirneisen et al 2010, Comprehensive Reviews in Food Science and Food Safety).

This work is quite interesting; however, in its present form it does not contribute in a significant way to what has been already known about the impact of ozone on calicivirus infectivity.

I would appreciate if the authors could address my following concerns:

1.     The “hermetic box for gas flow” that the authors used in this study: this device does not seem translatable by any means to a more practical and impactful application in disinfection of larger objects or (ideally) entire rooms in animal nosocomial settings and catteries to control FCV. Did the authors consider using an ozone generator placed in an isolated environment to assess the efficacy of gaseous ozone on FCV inactivation?

2.     The efficacy of the gaseous ozone treatment: despite using a small, hermetic device, where the gas input was in close proximity to the Petri dish containing the cells, the virucidal treatment (even after 10 minutes) led to a maximum of 2.25 log reduction in viral titer. I would expect a much higher efficacy in neutralizing FCV. Under non-cytotoxic conditions (Fig. 3 – antiviral assay), only up to 0.5 log FCV titer reduction with 20ug/ml or 35 ug/ml ozone treatment for 90s was achieved. This is a very modest, negligible reduction. In Fig. 4 (virucidal assay), the authors show up to 1 log reduction with the same concentration of ozone and duration of treatment. How do the authors explain this variability? It seems the 2 experiments in Fig. 3 and 4 were conducted in the same exact way (also, Fig. 4 seems to provide redundant information, for the sake of clarity it would be useful to exclude any treatment shorter than 90s from that figure, since it’s already present in Fig. 3).

3.     Compared to HuNoVs, FCV is more sensitive to low pH, to chlorine exposure, to surface drying, as well as to UVC irradiation (Ohmine et al. 2018, Biocontrol Science; Dultree et al. 1999, Journal of Hospital Infection; Rockey et al. 2020, Environmental Science & Technology). Poschetto et al. 2007 (Applied and Environmental Microbiology) tested FCV resistance to chemical disinfection, and concluded that “Generally NV appeared more resistant than FCV, and consequently, the suitability of FCV as a model for NV should be considered with caution”. Nevertheless, the authors didn’t acknowledge the well-known inferior stability and resistance to disinfection of FCV compared to HuNoV, despite concluding that their data open new perspectives in future applications of ozonation in human and veterinary medicine settings.

4.     There is no comparison whatsoever of the results of this study to results published by other authors on the efficacy of ozonation on virus inactivation. For example, in the recently published paper by Vojtkovská et al. (2023) in the journal Animals, the authors assessed the efficacy of gaseous ozone on 4 different enveloped and nonenveloped viruses (including FCV) in vitro, by using two different ozone generators. Vojtkovská et al didn’t assess viral titers by TCID50, but only the presence or absence of cytopathic effect in the same cell line used in this study, observing no FCV cytopathic effect after 6h of exposure to ozone (ozone generator generating 20g/h of ozone, placed 150 to 230 cm from the cells). Any treatment less than 6h was not effective against the virus (CPE was present). Considering that, what is the output (g of ozone per hour) of the device used in your study compared to the devices used in the study by Vojtkovská et al.? In summary, how do your results compare to those previously published that tested the efficacy of gaseous ozone on inactivation of FCV and other similar viruses?

Author Response

Dear Referee,

herein you can find a point-by-point response to your comments for the manuscript animals-2826879 entitled “Activity of Ozone against Feline Calicivirus” submitted to Animals.

Thanks in advance

Best regards

Gianvito Lanave

FCV is a common highly contagious virus affecting cats, and as other viruses of the Caliciviridae family, is known to be resistant to many disinfectants. Catella et al. examined the impact of gaseous ozone on FCV viability in vitro using a hermetic box for gas flow. The effect of gaseous ozone on disinfection of noroviruses has been previously studied, for example in the context of hospital settings (Hudson et al 2007, Journal of Hospital Infections) or food disinfection (Predmore et al 2015, Food Microbiology; Hirneisen et al 2010, Comprehensive Reviews in Food Science and Food Safety). This work is quite interesting; however, in its present form it does not contribute in a significant way to what has been already known about the impact of ozone on calicivirus infectivity.

I would appreciate if the authors could address my following concerns:

R.3.1. The “hermetic box for gas flow” that the authors used in this study: this device does not seem translatable by any means to a more practical and impactful application in disinfection of larger objects or (ideally) entire rooms in animal nosocomial settings and catteries to control FCV. Did the authors consider using an ozone generator placed in an isolated environment to assess the efficacy of gaseous ozone on FCV inactivation?

Reply to R.3.1. Hermetic box for gas flow has been previously described (Lillo et al., 2023, https://doi.org/10.3390/ani13121920). This box was adopted in our study to evaluate the cytotoxicity on confluent monolayers of CRFK cells and in vitro antiviral/virucidal activity of ozone against FCV. We agree with the referee that our experimental settings differ from what can be observed in “practical” settings. However, the hermetic box has the advantage of establishing reproducible and measurable conditions, with a cost-effective approach. We would like to point out that this reproducible, more scientific approach is also inspired by the work of Fontes et al., 2012 (Fontes, B., Cattani Heimbecker, A.M., de Souza Brito, G. et al. Effect of low-dose gaseous ozone on pathogenic bacteria. BMC Infect Dis 12, 358 (2012). https://doi.org/10.1186/1471-2334-12-358). We clearly disclosed this limitation of the study in the discussion at page 10, line 388-396.

R.3.2. The efficacy of the gaseous ozone treatment: despite using a small, hermetic device, where the gas input was in close proximity to the Petri dish containing the cells, the virucidal treatment (even after 10 minutes) led to a maximum of 2.25 log reduction in viral titer. I would expect a much higher efficacy in neutralizing FCV. Under non-cytotoxic conditions (Fig. 3 – antiviral assay), only up to 0.5 log FCV titer reduction with 20ug/ml or 35 ug/ml ozone treatment for 90s was achieved. This is a very modest, negligible reduction. In Fig. 4 (virucidal assay), the authors show up to 1 log reduction with the same concentration of ozone and duration of treatment. How do the authors explain this variability? It seems the 2 experiments in Fig. 3 and 4 were conducted in the same exact way (also, Fig. 4 seems to provide redundant information, for the sake of clarity it would be useful to exclude any treatment shorter than 90s from that figure, since it’s already present in Fig. 3).

Reply to R.3.2. We agree with the referee R3 that we were not clear in some points of the manuscript.

In the study we assessed antiviral and virucidal activity of ozone against FCV. The designed experiments included treatment of cells with O3 before/after the addition of FCV onto CRFK cells (antiviral effects), and treatment of virus before infection of cell monolayers (virucidal activity). Different O3 concentrations (20, 35 and 50 μg/ml) for different time intervals (30, 60 and 90s) were assessed, thus observing a significant decline of viral titer, as much as 1.75 log10, in protocol A (treatment with O3 of FCV-infected monolayers) at 50 μg/ml for 90s.

Moreover, we investigated the virucidal activity of O3 by treating the virus before cell adsorption, in contact with ozone at different concentrations (20, 35 and 50 μg/mL) for different time intervals (30, 60, 90, 180, 300 and 600 s). Based on the results, virucidal activity of O3 against FCV occurred in a dose- and time-contact-dependent fashion with the highest significant decline of viral titer, as much as 2.25 log10, observed using O3 at 50 μg/mL for 600s.

Different protocols were designed to assess the antiviral (figure 3) and virucidal (figure 4) activity of ozone.

Figure 3 shows the antiviral activity assessed with protocol A and B, plus the control virus at different times of exposure (panel A=30s, panel B=60s, panel C=90s). We added a short sentence to the legend of figure 3 “Antiviral activity of ozone (O3) at different concentrations...” (lines 263 page 7)

In figure 4 legend, we added a short sentence: “Virucidal activity of ozone (O3) at different concentrations….” (line 305 page 8)

R.3.3. Compared to HuNoVs, FCV is more sensitive to low pH, to chlorine exposure, to surface drying, as well as to UVC irradiation (Ohmine et al. 2018, Biocontrol Science; Dultree et al. 1999, Journal of Hospital Infection; Rockey et al. 2020, Environmental Science & Technology). Poschetto et al. 2007 (Applied and Environmental Microbiology) tested FCV resistance to chemical disinfection, and concluded that “Generally NV appeared more resistant than FCV, and consequently, the suitability of FCV as a model for NV should be considered with caution”. Nevertheless, the authors didn’t acknowledge the well-known inferior stability and resistance to disinfection of FCV compared to HuNoV, despite concluding that their data open new perspectives in future applications of ozonation in human and veterinary medicine settings.

Reply to R.3.3. The referee R3 mentions some examples of the vast existing literature to point out the FCV is less resistant than human NoVs. Indeed, the literature on this topic is quite intricate. For instance, in a paper by Di Martino et al, (2020, https://doi.org/10.1111/tbed.13605.), it seems that FCV exists as different biotypes. Enteric FCV strains are more resistant to pH and to trypsin than respiratory FCV isolates. So, there are so many variables and making general assumptions can be hazardous. Also, the study of Ohmine et al, 2018 tested FCV as surrogate of human NoV, in comparison with murine NoV, poliovirus and coxackievirus, since human NoVs are not cultivatable. Therefore, in this study the comparison was with a murine NoV and not with a human NoV.

Finally, the study of Rockey et al 2020 describes an alternative tool (based on the estimation of genome integrity) to viability PCR for the estimation of viable virus after UV exposure. Both the approaches indirectly estimate viable virus, since human NoV is not cultivatable. However, more precise information on human NoV properties is being generated using enteroid cells (Chan et al., 2019 https://doi.org/10.3201/eid2509.190205; Zou et al., 2019 https://doi.org/10.1007/7651_2017_1). Surprisingly, marked differences in terms of virus resistance to chemical and physical inactivation have been observed among the various NoV genotypes and even among GII.4 variants (Recker and Li, 2020 https://doi.org/10.4315/0362-028X.JFP-19-410; Park et al., 2016, https://doi.org/10.1371/journal.pone.0157787; Liu et al., 2011, https://doi.org/10.1007/s12560-011-9053-x; Li et al., 2012, https://doi.org/10.1016/j.jviromet.2012.01.001.

Taking into account the referee R3’s comments, we added in the discussion a sentence to disclose that “Our study has several limitations. Other studies should assess the in vitro broad-spectrum antiviral and virucidal efficacy of O3, taking into account the genetic and phenotypic di-versity of enteric viruses even below the species level. For instance, marked difference in terms of resistance to chemical and physical inactivation has been observed between NoV GII.4 variants [45-48]. Likewise, marked difference in terms of resistance phenotype have been observed between enteric and respiratory FCV isolates [49]. Furthermore, research studies are needed to validate the effectiveness of O3 in practical scenarios such as cat-teries and shelters. (see lines 388-396, page 10)

R.3.4. There is no comparison whatsoever of the results of this study to results published by other authors on the efficacy of ozonation on virus inactivation. For example, in the recently published paper by Vojtkovská et al. (2023) in the journal Animals, the authors assessed the efficacy of gaseous ozone on 4 different enveloped and nonenveloped viruses (including FCV) in vitro, by using two different ozone generators. Vojtkovská et al didn’t assess viral titers by TCID50, but only the presence or absence of cytopathic effect in the same cell line used in this study, observing no FCV cytopathic effect after 6h of exposure to ozone (ozone generator generating 20g/h of ozone, placed 150 to 230 cm from the cells). Any treatment less than 6h was not effective against the virus (CPE was present). Considering that, what is the output (g of ozone per hour) of the device used in your study compared to the devices used in the study by Vojtkovská et al.? In summary, how do your results compare to those previously published that tested the efficacy of gaseous ozone on inactivation of FCV and other similar viruses?

Reply to R.3.4. We cited the manuscript published by Vojtkovská et al. (2023) (see ref 15 in the revised manuscript) even if it was difficult for us to compare the results, since the experiments by Vojtkovská were based only on observation of cytopathic effect and the Authors did not disclose the PCR assays used to confirm the results (i.e, it is not mentioned if they used quantitative or qualitative assays and which protocols; the criteria they adopted with the molecular assays to assess virus replication are not disclosed). So, it is difficult for us to compare their findings.